# Circular oligomerization is an intrinsic property of synaptotagmin

Jing Wang[1†], Feng Li[1†], Oscar D Bello[1,2], Charles Vaughn Sindelar[3], Frédéric Pincet[1,4], Shyam S Krishnakumar[1,2*], James E Rothman[1,2*]

[1]Departments of Cell Biology, Yale University School of Medicine, New Haven, United States; [2]Department of Clinical and Experimental Epilepsy, Institute of Neurology, University College London, London, United Kingdom; [3]Departments of Molecular Biophysics and Biochemistry, Yale University School of Medicine, New Haven, United States; [4]Laboratoire de Physique Statistique, UMR CNRS 8550 Associée aux Universités Paris 6 et Paris 7, Paris, France

**Abstract** Previously, we showed that synaptotagmin1 (Syt1) forms $Ca^{2+}$-sensitive ring-like oligomers on membranes containing acidic lipids and proposed a potential role in regulating neurotransmitter release (*Zanetti et al., 2016*). Here, we report that Syt1 assembles into similar ring-like oligomers in solution when triggered by naturally occurring polyphosphates (PIP2 and ATP) and magnesium ions ($Mg^{2+}$). These soluble Syt1 rings were observed by electron microscopy and independently demonstrated and quantified using fluorescence correlation spectroscopy. Oligomerization is triggered when polyphosphates bind to the polylysine patch in C2B domain and is stabilized by $Mg^{2+}$, which neutralizes the $Ca^{2+}$-binding aspartic acids that likely contribute to the C2B interface in the oligomer. Overall, our data show that ring-like polymerization is an intrinsic property of Syt1 with reasonable affinity that can be triggered by the vesicle docking C2B-PIP2 interaction and raise the possibility that Syt1 rings could pre-form on the synaptic vesicle to facilitate docking.

DOI: https://doi.org/10.7554/eLife.27441.001

**\*For correspondence:**
shyam.krishnakumar@yale.edu (SSK);
james.rothman@yale.edu (JER)

[†]These authors contributed equally to this work

**Competing interests:** The authors declare that no competing interests exist.

## Introduction

Synaptic vesicle (SV) fusion at the neuronal synapse is mediated by the SNARE (soluble NSF attachment protein receptor) complex, whose assembly is chaperoned by several other proteins to achieve precision and synchronicity of neurotransmitter release (*Südhof, 2013*; *Söllner et al., 1993a*, *1993b*). These include Munc18, Munc13, complexin, synaptotagmin1 (Syt1), α-SNAP, and NSF (*Südhof, 2013*; *Südhof and Rothman, 2009*; *Jahn and Fasshauer, 2012*; *Rizo and Xu, 2015*). SV-associated protein Syt1 is of special interest because it is the primary calcium ($Ca^{2+}$) sensor that triggers the rapid, synchronous neurotransmitter release after $Ca^{2+}$ are admitted into the pre-synaptic terminal following an action potential (*Brose et al., 1992*; *Fernández-Chacón et al., 2001*; *Geppert et al., 1994*). It is well-established that the $Ca^{2+}$-induced reorientation of the cytoplasmic C2 domains (C2A and C2B) of Syt1 into the membrane is driving force behind this process (*Rhee et al., 2005*; *Paddock et al., 2011*; *Hui et al., 2006*; *Krishnakumar et al., 2013*), but the precise molecular details are still unclear. Under resting conditions, Syt1 is also involved in pre-fusion docking and priming of SVs to the plasma membrane (PM). This involves the interaction of the polybasic patch of C2B domain with the negatively charged phospholipids, like phosphatidylinositol 4,5-bisphosphate (PIP2) and phosphatidylserine (PS) in the PM (*Wang et al., 2011*; *Bai et al., 2004*; *Honigmann et al., 2013*; *Parisotto et al., 2012*; *Pérez-Lara et al., 2016*; *Martin, 2015*), and independent but concurrent binding to the t-SNAREs, Syntaxin and SNAP-25 (*Zhou et al., 2015*; *de Wit et al., 2009*; *Kedar et al., 2015*; *Mohrmann et al., 2013*). Also, multiple lines of evidence (*Hua and*

*Charlton, 1999*; *Walter et al., 2010*; *Li et al., 2014*; *Giraudo et al., 2006*) imply that prior to the entry of $Ca^{2+}$, the SNARE complex is approximately half-zippered but unable to complete its zippering to drive fusion and Syt1, along with Complexin, might be involved in maintaining this pre-fusion activated state. Yet, the molecular mechanism by which Syt1 prevents such half-zippered SNAREpins from completing before $Ca^{2+}$ entry and the mechanisms by which Syt1 can release or activate (or a combination) fusion after $Ca^{2+}$ entry are unknown.

Recently, we described a novel organization of Syt1 on membranes which could mechanistically explain the multiple roles of Syt1 in orchestrating synchronous neurotransmitter release (*Wang et al., 2014*; *Zanetti et al., 2016*). We found that Syt1 self-assembles into ~30 nm sized rings (~17 copies of Syt1) on lipid surfaces, which disassemble upon binding $Ca^{2+}$ in the physiologically-relevant range (*Wang et al., 2014*; *Zanetti et al., 2016*). This suggests a simple physical mechanism for regulation of synaptic exocytosis. While no doubt over-simplified, it is easy to imagine how an interposed protein ring at the docked SV-PM interface would act as a reversible barrier to fusion that is blocking SV fusion until it is disassembled by $Ca^{2+}$ influx thereby helping to synchronize neurotransmitter release. Such a ring might further serve as platform to organize multiple SNAREpins to act cooperatively to open the fusion pore faster.

While there is as yet no direct evidence that such rings exist in vivo at the SV-PM junction, nonetheless striking genetic and biochemical correlations provided by the ring structure (*Wang et al., 2014*; *Zanetti et al., 2016*) and the fact that the ring-like oligomers are a conserved structural feature of many C2 domain proteins (*Zanetti et al., 2016*) strongly support the overall hypothesis. Building upon our previous work (*Zanetti et al., 2016*), and our recent finding that Syt1 oligomers can be triggered in solution by polyphosphates and $Mg^{2+}$. In this research advance, we use fluorescence correlation spectroscopy to independently corroborate the Syt1 oligomeric structure and provide mechanistic insight into the Syt1 ring oligomer assembly.

## Results

### PIP2 or ATP and Mg²⁺ promote Syt1 ring oligomers in solution

The minimal C2AB domains of Syt1 form circular oligomers on lipid surface and the electrostatic interaction between the conserved lysine residues (K326/K327) within the polybasic patch/motif on the C2B domain and the negatively-charged lipids, like PIP2 and PS, on the lipid surface was required for its assembly (*Wang et al., 2014*; *Zanetti et al., 2016*). To understand if this electrostatic interaction in itself is the trigger to polymerize or if it merely serves to position the Syt1 on the membrane to promote the ring formation, we examined if the Syt1 rings could be assembled in solution using acidic lipid substitutes. We incubated stringently purified C2AB domains of Syt1 (Syt1$^{C2AB}$) with soluble PIP2 (PIP2-diC4) under physiologically-relevant buffer condition (100 mM KCl, 1 mM free $Mg^{2+}$) and imaged the resultant structures using electron microscopy (EM). Negative stain analysis showed that in the presence of 50 μM PIP2-diC4, Syt1$^{C2AB}$ readily assembles into ring-like oligomers in solution (*Figure 1A*). The density (~5 rings/ μm²) and the dimension (outer diameter between 18–44 nm with an average size of ~32 nm) of these soluble ring oligomers were very similar to the ring oligomers formed on the lipid surface (*Figure 1B and C*). Based on the helical indexing of the Syt1$^{C2AB}$ tubes (*Wang et al., 2014*), we estimate that this corresponds to 12–26 copies of Syt1 molecule, with an average of ~17 copies of Syt1. Notably, the soluble Syt1$^{C2AB}$ rings are abundant and stable under physiological buffer conditions, unlike the sparse density observed on lipid monolayers (*Wang et al., 2014*). This suggests that circular oligomerization is an intrinsic property of Syt1 C2 domains, but hindered by the low concentration of the protein on the lipid monolayer surface under these conditions.

Similar ring oligomers were observed with PIP2 analogues like inositol 1,4,5-trisphosphate (IP3, 50 μM) (*Figure 1D*) suggesting it was not a unique property of PIP2. Consequently, we tested the effect of adenosine triphosphate (ATP), another polyvalent anion, which has been shown to directly bind the C2B polybasic region and modulate the membrane interaction of Syt1 (*Park et al., 2012*; *Vennekate et al., 2012*). As with soluble PIP2 and its analogues, we detected ring-like oligomers of Syt1$^{C2AB}$ (*Figure 1E*) at physiological concentration of ATP (1 mM Mg-ATP), with density (~5 rings/μ

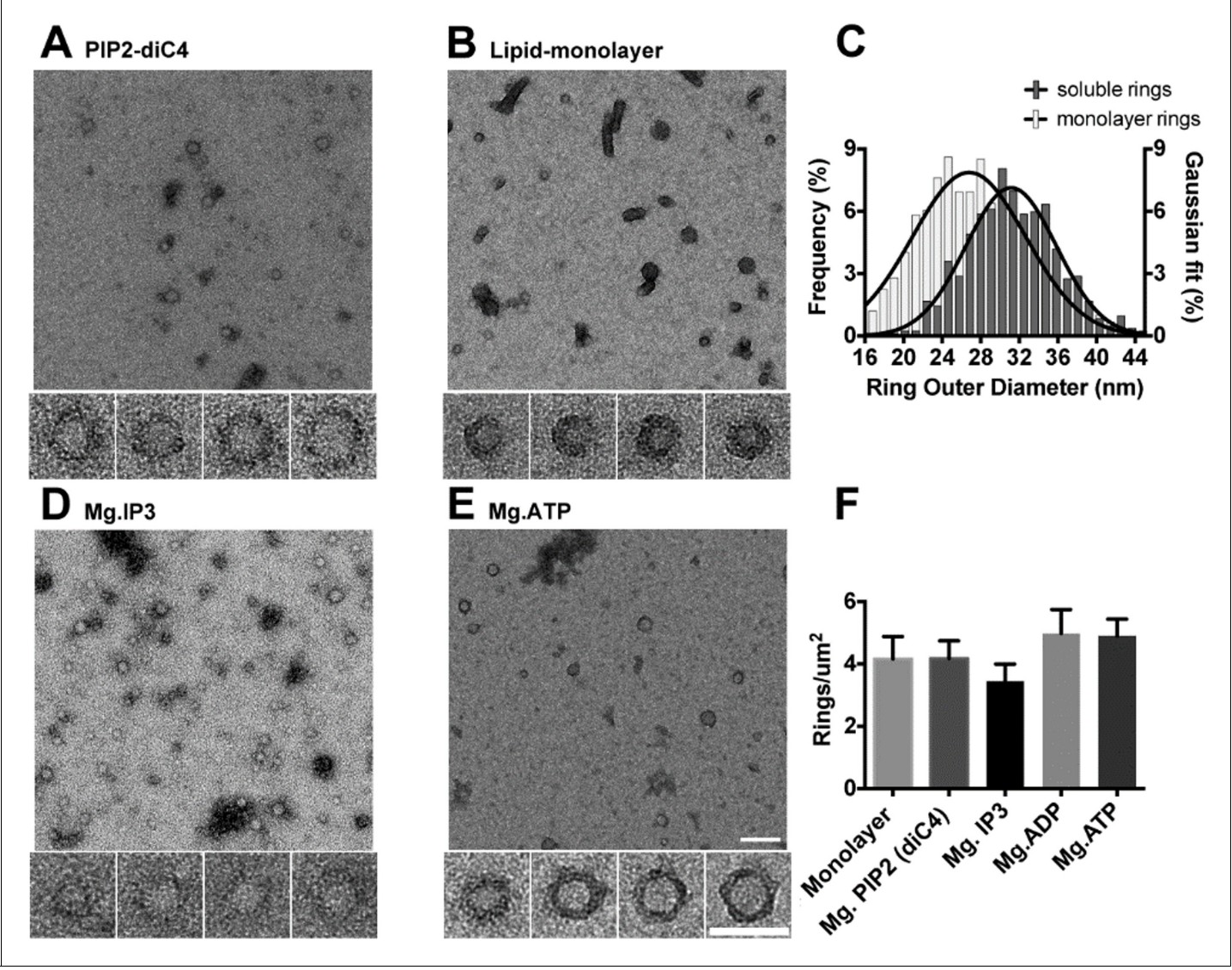

**Figure 1.** The C2AB domains of Syt1 oligomerize into ring-like structures in solution when triggered with naturally occurring polyphosphates and $Mg^{2+}$. Negative staining electron microscopy analysis shows that $Syt1^{C2AB}$ forms ring-like structures in buffer solution containing soluble PIP2 (PIP2-diC4) and 1 mM free $Mg^{2+}$ (**A**), and these rings are very similar to those observed on monolayers containing 40% DOPS (**B**). (**C**) The average diameter of the $Syt1^{C2AB}$ soluble ring-oligomers are slightly but not significantly larger (32 ± 3 nm) compared to the rings (28 ± 2 nm) formed on lipid monolayer. These oligomeric rings were not unique to PIP2-diC4 and other polyvalent anions, like IP3 (**D**) and ATP (**E**) also induced the oligomerization of $Syt1^{C2AB}$. The prevalence and stability of the $Syt1^{C2AB}$ were similar for all the polyvalent anions tested (**F**). The scale bar represents 100 nm in the main figure and 50 nm for the insets (Panels **A, B, D, E**). Average values and standard deviations from a minimum of 3 independent experiments are shown.
DOI: https://doi.org/10.7554/eLife.27441.002

$m^2$) comparable to soluble PIP2. Similar ring oligomers were also observed with 1 mM adenosine diphosphate (ADP) suggesting that it is a shared property of polyvalent anions (*Figure 1F*).

No oligomeric structures were observed in the absence of polyvalent anions and the density of the rings was drastically reduced (~5 rings/$\mu m^2$ to ~0.4 rings/$\mu m^2$) when the polybasic region of C2B was mutated (K326A/K327A) (*Figure 2A*). This suggests that the known direct molecular interaction of polyanions with the C2B polybasic motif (*Park et al., 2012*; *Bai et al., 2004*) and not a charge shielding effect, produces the Syt1 ring-like oligomers. Magnesium ions ($Mg^{2+}$) were also found to be an important co-factor in stabilizing the $Syt1^{C2AB}$ rings in solution (*Figure 2B and C*). The number of $Syt1^{C2AB}$ ring oligomers observed were substantially reduced (~60% reduction) when $Mg^{2+}$ was completely excluded (using Na-ATP or Na-PIP2) (*Figure 2C*). Given that the $Mg^{2+}$ was not required

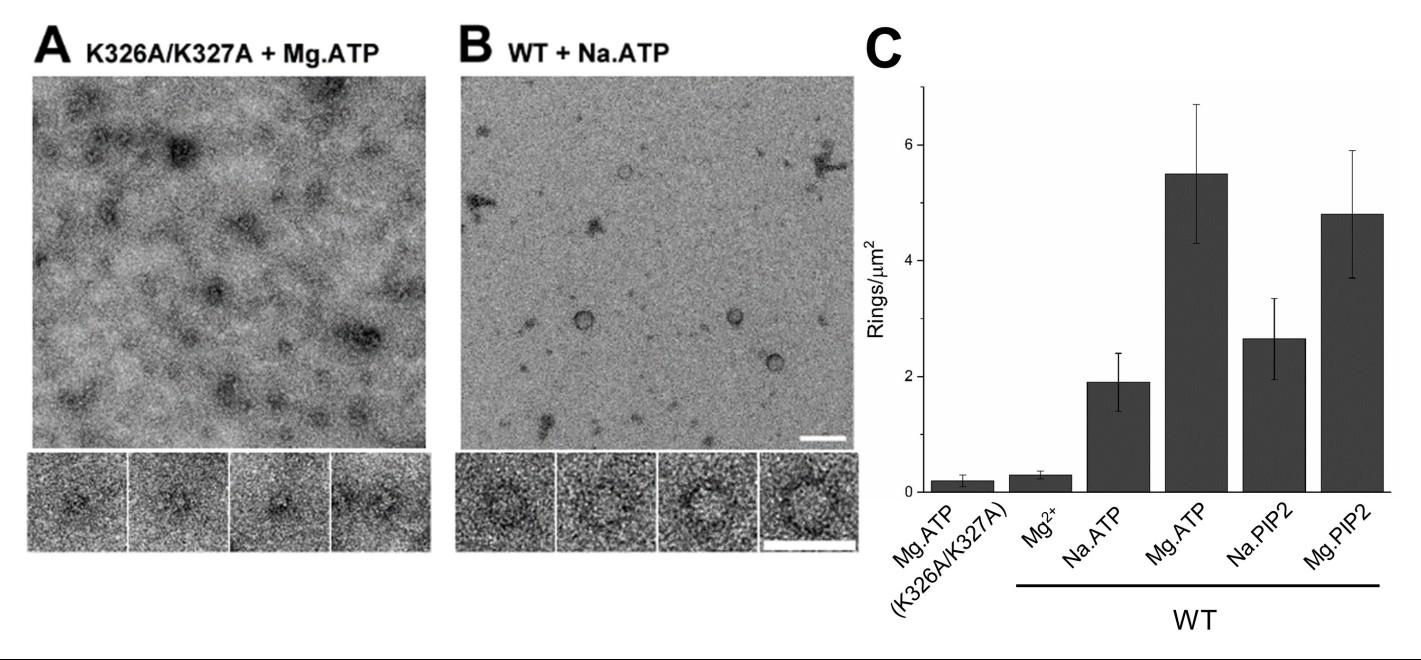

**Figure 2.** (A) Molecular interaction of polyvalent anion with the C2B polybasic is strictly required for the oligomer formation. Negative stain EM analysis shows that disrupting the polylysine motif on the C2B domain (K326A/K327A) prevents the oligomer formation even in the presence of polyphosphates and $Mg^{2+}$. (B and C) $Mg^{2+}$ is a critical co-factor that stabilizes the ring oligomers triggered by polyphosphates (ATP or PIP2) as complete removal of $Mg^{2+}$ (Na.ATP or Na.PIP2) results in reduction in the density of the ring oligomers.

DOI: https://doi.org/10.7554/eLife.27441.003

The following figure supplement is available for figure 2:

**Figure supplement 1.** Glycerol density gradient was used to purify the soluble Syt1$^{C2AB}$ ring oligomers.

DOI: https://doi.org/10.7554/eLife.27441.004

to assemble the Syt1$^{C2AB}$ ring oligomers on acidic lipid surfaces (in fact $Mg^{2+}$ lowered the number of rings formed on lipid surfaces), it appears that $Mg^{2+}$ plays an independent but auxiliary role in the formation of Syt1$^{C2AB}$ ring oligomers in solution. We also attempted to isolate the soluble Syt1$^{C2AB}$ rings (triggered with ATP and $Mg^{2+}$) using glycerol density gradient. However, the soluble rings were not stable under these experimental conditions and could be isolated only after mild fixation (0.01% glutaraldehyde) (*Figure 2—figure supplement 1*). Taken together, our data suggest that the specific interaction of the polyvalent anions to the conserved lysine residues within the C2B polybasic motif triggers the Syt1 ring-like oligomer formation and the resultant oligomers are stabilized by $Mg^{2+}$.

## Characterization of soluble Syt1 oligomers by Fluorescence Correlation Spectroscopy

To independently validate and characterize Syt1$^{C2AB}$ oligomerization in solution, we used fluorescence correlation spectroscopy (FCS). FCS is based on correlation analysis of temporal fluctuations of fluorescence intensity caused by diffusion of fluorescently labeled moiety through a small focal volume (dimension less than 1 pL). Autocorrelation of the fluorescence can directly provide the average number of particles (concentration) and average diffusion times through the volume. And, cross-correlation analysis between two fluorophores can be used to detect molecular association/dissociation and to determine the stoichiometry of molecular complexes (*Elson, 2011*; *Magde et al., 1974*; *Ries and Schwille, 2012*). So, to check the oligomerization of Syt1$^{C2AB}$ in solution, we employed dual-color fluorescence cross-correlation spectroscopy (FCCS) with Syt1$^{C2AB}$ labeled with Alexa488 or Alexa647 at residue 269 using cysteine-maleimide chemistry. The fluorescent labels were introduced at the flexible linker region between C2A and C2B domains and negative stain EM analysis

showed the fluorescent labels did not interfere the Syt1 ring formation (*Figure 3—figure supplement 1*).

FCCS analysis was carried out under experimental conditions similar to the EM analysis, namely 5 µM of Syt1$^{C2AB}$ (~120 nM of Syt1$^{C2AB}$-Alexa88, ~30–120 nM of Syt1$^{C2AB}$-Alexa647 and ~4.9 µM of unlabeled Syt1$^{C2AB}$) was incubated with or without ATP in buffers containing 20 mM KCl and 1 mM free Mg$^{2+}$. Both fluorophores were excited, and their intensities in the focal volume were monitored simultaneously. In the presence of ATP and Mg$^{2+}$, the temporal variations in the fluorescence intensity of Alexa488 and Alexa647 labels followed a similar pattern, indicating that part of Syt1$^{C2AB}$-Alexa488 and Syt1$^{C2AB}$-Alexa647 molecules move in or out the focal volume concurrently (*Figure 3A*). Also, the cross-correlation function was larger than one at short time intervals (*Figure 3A*), indicating that a portion of the two fluorophores diffused together that is, they were in the same diffusional object. Overall, these data were consistent with the formation of Syt1$^{C2AB}$ oligomers in solution. Further, we observed no cross correlation (*Figure 3B*) in the absence of ATP and Mg$^{2+}$, implying that these structures are not random association of Syt1$^{C2AB}$ molecules, but are ATP-derived oligomers of Syt1$^{C2AB}$.

To corroborate these findings, we estimated the diffusion properties of the fluorescent species. Higher molecular weight of the Syt1$^{C2AB}$ oligomers would increase the diffusion time (transit time) across the sample volume, and this could be determined by autocorrelation analysis. We used both fluorophores to cross-check our results, but lowered the concentrations of each fluorescent species (~25 nM of Syt1$^{C2AB}$-Alexa488, ~25 nM of Syt1$^{C2AB}$-Alexa647 and ~5 µM of unlabeled Syt1$^{C2AB}$), such that two fluorophores are unlikely to be located in the same oligomer. This was confirmed by the absence of cross correlation between the two fluorophores (*Figure 3—figure supplement 2*). We then measured the auto correlation function of each fluorophore in presence of ATP and Mg$^{2+}$. We found that they could be best fitted using the two-component translational diffusion model with a triplet state correction (*Equation 1*) and determined two diffusion coefficients corresponding to the Syt1$^{C2AB}$ monomers and oligomers.

$$G(\tau) = 1 + A\left(1 + \frac{\phi_{tri}e^{-\frac{\tau}{\tau_{tri}}}}{1 - \phi_{tri}}\right)\left(\frac{\phi_m}{\left(1 + \frac{\tau}{\tau_m}\right)\left(1 + \frac{1}{s^2}\frac{\tau}{\tau_m}\right)^{1/2}} + \frac{\phi_o}{\left(1 + \frac{\tau}{\phi_o}\right)\left(1 + \frac{1}{s^2}\frac{\tau}{\phi_o}\right)^{1/2}}\right) \tag{1}$$

where G is the auto correlation function, $\tau$ is the time interval, A is amplitude, $\Phi_{tri}$ and $\tau_{tri}$ are the fraction and decay time of the triplet state, $\Phi_m$ and $\tau_m$ are the fraction and diffusional time of the component 1, $\Phi_o$ and $\tau_o$ are the fraction and diffusional time of the component 2, and s is the structural parameter.

The major component (~80%) showed fast diffusion time (~600 µs) possibly corresponding to the monomers and a smaller fraction (~20%) had a slow diffusion time of ~4000 µs consistent with Syt1-$^{C2AB}$ oligomers (*Figure 3C, Table 1*). These data indicated that Syt1$^{C2AB}$ indeed formed oligomers in solution when triggered with ATP and Mg$^{2+}$. In contrast, in the absence of ATP and Mg$^{2+}$, the auto correlation function could be best fitted with the one component translational diffusion model with triplet state correction (*Equation 2*)

$$G(\tau) = 1 + A\left(1 + \frac{\phi_{tri}e^{-\frac{\tau}{\tau_{tri}}}}{1 - \phi_{tri}}\right)\frac{1}{\left(1 + \frac{\tau}{\tau_d}\right)\left(1 + \frac{1}{s^2}\frac{\tau}{\tau_d}\right)^{1/2}} \tag{2}$$

The diffusion time ($\tau_d$) obtained (~600 µs for both channels) were similar to the diffusion time of the monomeric component obtained in the two component diffusion model, confirming that no oligomeric structure is formed in the absence of Mg$^{2+}$ and ATP (*Table 1*).

To verify that the Syt1$^{C2AB}$ oligomerization observed in FCS analysis is indeed due to specific association of ATP to the C2B polybasic region, we tested the polylysine mutant (wild type Syt1$^{C2AB}$-Alexa647 and unlabeled Syt1$^{C2AB}$ K326A/K327A). For this mutant, in the presence of ATP and Mg$^{2+}$, the autocorrelation function was best described with one-component diffusion model with a diffusion time of 599 ± 50 µs and a diffusion coefficient of 72 ± 7 µm$^2$/s, which are similar to the property of a Syt1 monomer (*Figure 3—figure supplement 3*). These data are consistent with the EM analysis that the probability of forming Syt1 oligomeric structures is significantly decreased when the polylysine patch on C2B domain was altered. We also tested and confirmed that similar

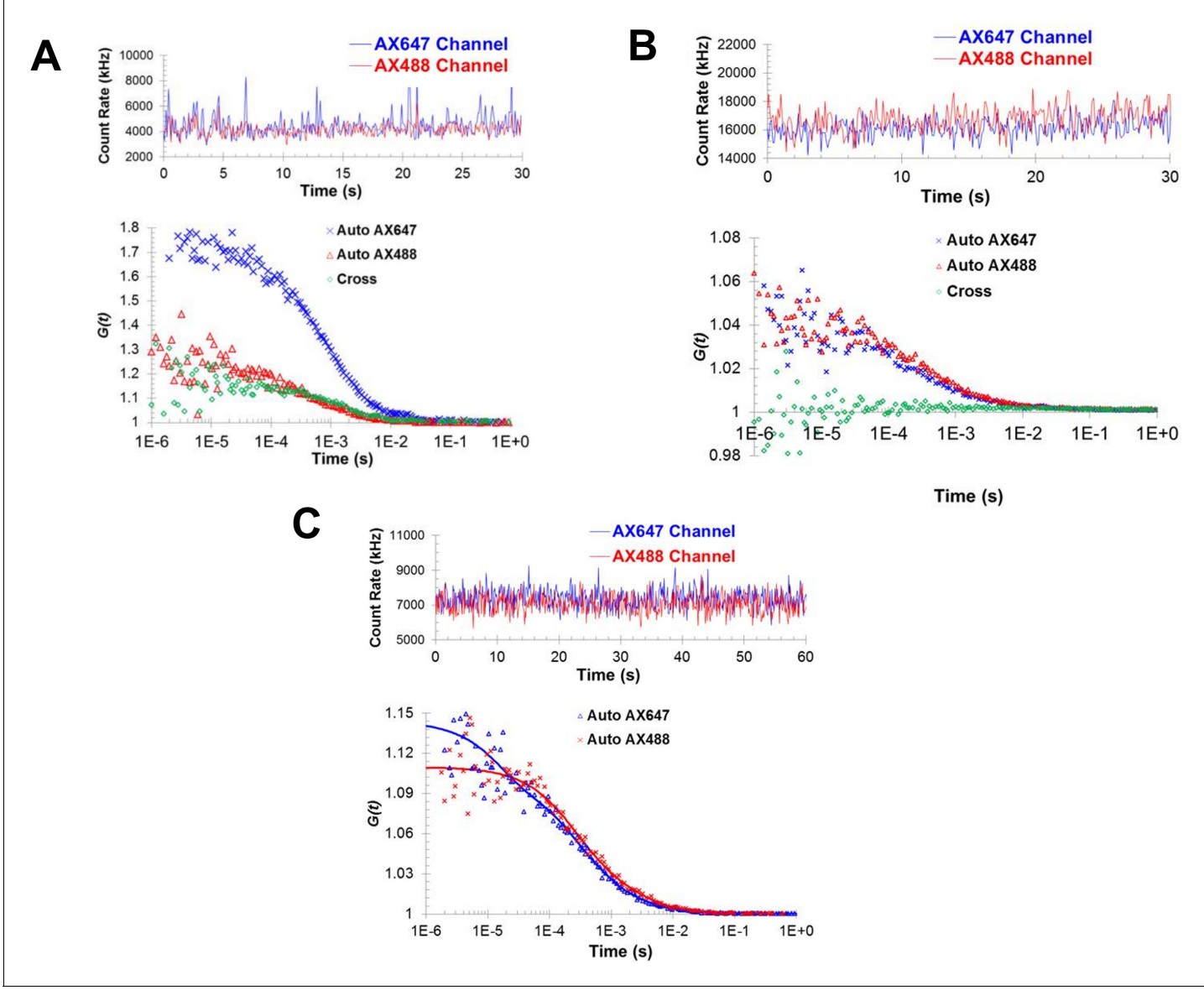

**Figure 3.** Fluorescence correlation spectroscopy confirms the formation of Syt1$^{C2AB}$ oligomers in solution. Fluorescence cross-correlation spectroscopy of the Syt1$^{C2AB}$ oligomerization reaction was measured using Alexa488 and Alexa647 labeled Syt1$^{C2AB}$, mixed with unlabeled Syt1$^{C2AB}$ protein either in the presence (**A**) or absence (**B**) of 1 mM ATP and Mg$^{2+}$. The upper panels represent the count rates and the lower panels represent the auto-correlation (blue and red markers for Alexa647 and Alexa488) and cross correlation (green markers) functions. The data were collected for a range of fluorophores concentration (~30–120 nM) and we observe no cross-correlation even at higher label concentration in the absence of Mg$^{2+}$/ATP (**B**) in contrast to distinct cross-correlation in the presence of Mg$^{2+}$/ATP (**A**) despite the lower label concentration. This shows that these structures are not random association of Syt1$^{C2AB}$ molecules, but are ATP-derived oligomers of Syt1C$^{2AB}$. (**C**) Fluorescence auto correlation spectroscopy of the oligomerization reaction was carried out using dilute concentration of the labeled Syt1$^{C2AB}$ (~25 nM each of Alexa488 and Alexa647) mixed with unlabeled Syt1$^{C2AB}$ in the presence of ATP and Mg$^{2+}$ The auto correlation function was best fitted with the two-component translational diffusion model with triplet state (blue and red solid lines for each dye), confirming the presence of the oligomeric component. Representative traces are shown and the calculated parameters from a minimum of three independent trials are shown in *Table 1*.

DOI: https://doi.org/10.7554/eLife.27441.005

The following figure supplements are available for figure 3:

**Figure supplement 1.** Negative stain EM analysis shows that the fluorescent label (Alexa488) introduced at position 269 does not affect Syt1$^{C2AB}$ ability to form ring oligomers on lipid monolayers containing 40% DOPS.

DOI: https://doi.org/10.7554/eLife.27441.006

**Figure supplement 2.** Fluorescent cross-correlation analysis confirms that under the low concentration of Alexa488 and Alexa647 used in the autocorrelation analysis, the two fluorophores are not located in the same oligomer and thus, could be used independently to cross-verify the analysis.

*Figure 3 continued on next page*

*Figure 3 continued*

DOI: https://doi.org/10.7554/eLife.27441.007

**Figure supplement 3.** Autocorrelation analysis of the oligomerization reaction of C2B polylysine motif mutant was carried out using the mixture of wild type Syt1$^{C2AB}$-Alexa647 and unlabeled Syt1 $^{C2AB}$ K326A/K327A.

DOI: https://doi.org/10.7554/eLife.27441.008

**Figure supplement 4.** Fluorescence cross-correlation and autocorrelation analysis shows the formation of Syt1$^{C2AB}$ oligomers with PIP2-diC4 and Mg$^{2+}$.

DOI: https://doi.org/10.7554/eLife.27441.009

**Figure supplement 5.** Geometry of the torus model (*Thaokar, 2008*) used to estimate the hydrodynamic radius of the Syt1$^{C2AB}$ oligomers.

DOI: https://doi.org/10.7554/eLife.27441.010

oligomers were formed with PIP2-diC4 and Mg$^{2+}$ using both autocorrelation (diffusion time) and cross-correlation analysis (*Figure 3—figure supplement 4*).

## Physical parameters of soluble Syt1 ring oligomers

To fully characterize the Syt1$^{C2AB}$ oligomers detected in the FCS analysis, we determined the key molecular parameters of Syt1 oligomers observed in the FCS lt. First, we estimated the average copy number of Syt1 in the oligomer from the FCS autocorrelation analysis. To do this, we considered the probability of a diffusional object in the focal volume to contain dye 1:

$$p = \frac{C_1}{C_{total}} \tag{3}$$

where $C_1$ and $C_{total}$ are the input concentrations of Syt1$^{C2AB}$–dye1 and all the Syt1$^{C2AB}$ (labeled +-unlabeled), respectively. Therefore, if an oligomer containing dye 2 has *N* copies of Syt1$^{C2AB}$ molecules, then the probability of it also containing dye 1 is

$$p_{12} = 1 - (1-p)^{N-1} \tag{4}$$

From the auto correlation functions, we can determine the concentration of oligomers that contain dye 2, which can be defined as $C_2 \phi_{2,o}$, where $\phi_{2,o}$ is the molar ratio of oligomers containing dye 2 versus all diffusional particles containing dye 2. Therefore, using $\phi_{2,o}$ and $C_2$, $p_{12}$ can be rewritten as

$$p_{12} = C_{12}/(C_2 \phi_{2,o}) \tag{5}$$

where $C_{12}$ is the concentration of diffusional objects that contain both Syt1-Alexa647 and Syt1-Alexa488 and can be calculated using

**Table 1.** Fractions and molecular properties of diffusional particles of Syt1$^{C2AB}$ generated various conditions as measured by fluorescence correlation spectroscopy.

Averages and standard errors from three independent trials are shown.

| Proteins | Total protein conc. (nM) | Mg$^{2+}$ + polyphosphate | Readout channel | Component 1 | | | Component 2 | | |
|---|---|---|---|---|---|---|---|---|---|
| | | | | Fraction (%) | Diffusion time (μs) | Diffusion coefficient (μm$^2$/s) | Fraction (%) | Diffusion time (μs) | Diffusion coefficient (μm$^2$/s) |
| Syt1-Alexa488 | ~30 | No | Alexa488 | 100 | 620 ± 35 | 70 ± 4 | n/a | n/a | n/a |
| Syt1-Alexa647 | ~30 | No | Alexa647 | 100 | 616 ± 22 | 70 ± 3 | n/a | n/a | n/a |
| Syt1-Alexa488 + Syt1-Alexa647 + unlabeled Syt1 | ~5000 | No | Alexa488 | 100 | 583 ± 20 | 74 ± 8 | n/a | n/a | n/a |
| | | | Alexa647 | 100 | 577 ± 20 | 75 ± 8 | n/a | n/a | n/a |
| Syt1-Alexa488 + Syt1-Alexa647 + unlabeled Syt1 | ~5000 | Yes | Alexa488 | 77 ± 2 | 527 ± 20 | 82 ± 4 | 23 ± 2 | 4168 ± 194 | 10 ± 1 |
| | | | Alexa647 | 76 ± 2 | 567 ± 26 | 77 ± 4 | 24 ± 2 | 3862 ± 196 | 11 ± 1 |

DOI: https://doi.org/10.7554/eLife.27441.011

$$C_{12} = \frac{(G_x(0)-1)\phi_{1,o}\phi_{2,o}}{(G_1(0)-1)(G_2(0)-1)V_{eff}} \tag{6}$$

Where $G_x(0)$ is the value of the cross-correlation function at zero time interval, $G_1(0)$ and $G_2(0)$ are the values of the auto correlation functions of dye 1 and 2 at zero time interval, respectively, and $V_{eff}$ is the effective focal volume. Combining *Equations 4–6*, we estimate the average copy number (N) of Syt1 in the oligomers to be ~16 ± 2. This perfectly matches with the estimated copies of Syt1 (~17 copies) in the ring-like oligomers from the EM analysis.

Next, we determined the hydrodynamic dimensions of the Syt1 monomeric and oligomeric component from its diffusion properties. For the monomers, we considered them as global structures and used Stokes-Einstein equation (*Equation 7*)

$$D = \frac{k_B T}{6\pi\eta r} \tag{7}$$

where $D$ is the diffusion coefficient determined from the autocorrelation function, $k_B$ is Boltzmann constant, $T$ is the temperature, $\eta$ is the viscosity of the solvent and $r$ is the hydrodynamic radius. For the oligomer, we used torus model (*Figure 3—figure supplement 5*) considering the ring-like oligomer structure, wherein the hydrodynamic radius can be determined from the following equation:

$$D = \frac{k_B T}{8\pi^2 \eta a}\left[ln\left(\frac{8a}{r}\right)+\frac{1}{2}\right] \tag{8}$$

$r$ is the radius of the sphere being rotated, and $a$ is the distance from the center of the circle to the axis of rotation (*Thaokar, 2008*). We estimate an average hydrodynamic radius of ~$r$ = 3.1±0.2 nm for the monomer and ~ $(r + a)$=24 ± 3 nm for Syt1 oligomer. Both of these values are in line with the dimension of Syt1$^{C2AB}$ monomer (~4.3 nm from X-ray crystallography [*Fuson et al., 2007*]) and ring oligomer (~32 nm from the EM analysis). Overall, the data suggest that Syt1 oligomers observed in FCS analysis are the same circular oligomers visualized by EM, given the molecular properties of both are perfectly aligned.

Finally, we used the FCS analysis to estimate the Syt1 oligomeric interaction. Because we observed two types of diffusional particles (Syt1 monomer and oligomer with average copy number N ~ 16) in the fluorescence correlation measurements, to determine the binding affinity, we considered a polymerization reaction that forms the oligomer ($Syt1_N$):

$$N\ .Syt1 \leftrightarrow Syt1_N$$

The binding affinity constant ($K_D$) can be written as

$$(K_D)^{N-1} = \frac{[Syt1]^N}{[Syt1_N]}$$

And the mass balance equation is,

$$[Syt1] + N.[Syt1_N] = [Syt1]_0$$

where $[Syt1]_0$ is the initial total input concentration of Syt1.

The molar fractions of the fluorescent dyes in the monomers ($\Phi_{1,m}$ and $\Phi_{2,m}$) were determined from the auto correlation functions of the Alexa647 and Alexa488 channels, respectively. As shown in *Table 1*, the two dyes, Syt1-Alexa647 (dye 1) and Syt1-Alexa488 (dye 2), have similar distribution in monomers and oligomers, that is $\Phi_{1,m} \cong \Phi_{2,m}$, where $\Phi_{1,m}$ is the molar fraction of dye 1 in the form of monomer. Therefore, we can reasonably assume that the unlabeled Syt1 molecules have same distribution as Syt1-Alexa647 and Syt1-Alexa488. Then,

$$\phi_{1,m} \cong \phi_{2,m} \cong \frac{[Syt1]}{[Syt1]+[Syt1_N]} \tag{9}$$

Combining these equations together, we can write the $K_D$ as

$$K_{\mathrm{D}} = \frac{\phi_{1,m}^{\frac{1}{(N-1)}}}{\left(1-\phi_{1,m}\right)^{\frac{1}{(N-1)}}\left(N\left(1-\phi_{1,m}\right)+\phi_{1,m}\right)}[Syt1]_0 \tag{10}$$

The values of $\Phi_{1,m}$ and $\Phi_{2,m}$ were in good agreement and found to be ~0.77 (*Table 1*). From *Equation 10*, the affinity constant ($K_D$) for the Syt1$^{C2AB}$ monomers to assemble into the oligomers was estimated to be ~1 ± 0.3 μM. This corresponds to a free energy of ~−13.8 $k_B$T for a monomer and ~−220 $k_B$T for the oligomer with 16 copies of Syt1$^{C2AB}$. This suggests that Syt1 ring oligomers could reasonably assemble under the physiological conditions when triggered by naturally occurring polyphosphates.

## Syt1 soluble rings are insensitive to calcium

Syt1$^{C2AB}$ ring oligomers assembled on the membrane surface are sensitive to $Ca^{2+}$ and are rapidly disrupted by $Ca^{2+}$ wash at physiological concentrations (*Wang et al., 2014*; *Zanetti et al., 2016*). This sensitivity maps to the $Ca^{2+}$ binding to the C2B domain and the subsequent insertion of the flanking aliphatic residues into the membrane (*Zanetti et al., 2016*). In line with this, we find that the soluble Syt1$^{C2AB}$ rings are insensitive to $Ca^{2+}$ treatment (*Figure 4*). Negative stain EM analysis showed the Syt1$^{C2AB}$ assembled into soluble ring-like oligomers in the presence of $Ca^{2+}$/ATP similar to those observed with ATP and $Mg^{2+}$ (*Figure 4A*). In fact, $Ca^{2+}$ was slightly better in stabilizing the ring oligomers across the concentration range (0.25–1 mM) tested (*Figure 4—figure supplement 1*). Similarly, we detected the presence of oligomeric structures with ATP and $Ca^{2+}$ in the FCS analysis, and the diffusional properties of the oligomeric component was similar to that observed with ATP and $Mg^{2+}$ (*Figure 4B*). Hence, we conclude that the $Ca^{2+}$-induced membrane interaction of the C2B domain is essential for the dissociation of the Syt1 ring oligomers following the $Ca^{2+}$ influx. Remarkably, neutralizing the $Ca^{2+}$ coordinating aspartic acids of the C2B domain (Syt1$^{3A}$, D309A/D363A/D365A) stabilized the soluble ring oligomers. In fact, Syt1$^{3A}$ formed stable ring oligomers in the absence of divalent cations (with Na.ATP), with density comparable to those assembled with ATP and $Mg^{2+}$ (*Figure 4C*). This supports the idea that C2B $Ca^{2+}$ loops are involved the ring oligomer formation (*Wang et al., 2014*) and $Mg^{2+}$ stabilizes the ring oligomers by charge shielding effect that is by reducing the electrostatic repulsion between the Syt1 monomers by

## Discussion

The data presented here establish that circular oligomerization is an intrinsic property of Syt1, and it can assemble into stable ring-like oligomers independent of a membrane surface. However, this is not a spontaneous association and strictly requires the structurally defined binding of physiologically occurring polyphosphates anions like PIP2 or ATP to the polylysine motif on the C2B domain. In the case of PIP2, it is known that this interaction firmly docks the vesicle to the PM both under in vitro and in vivo conditions (*Honigmann et al., 2013*; *Parisotto et al., 2012*; *Pérez-Lara et al., 2016*). Structural analysis of the Syt1 ring oligomers from helical reconstruction of the Syt1$^{C2AB}$ coated membrane tubes (*Wang et al., 2014*) shows that the C2B calcium loops either contribute to or are in close proximity to the C2B-C2B interface that assembles the oligomers. Corroborating this, we find that neutralizing the $Ca^{2+}$-coordinating aspartic acids by alanine mutation or by inclusion of divalent cations ($Mg^{2+}$/$Ca^{2+}$) increase the prevalence and stability of the soluble Syt1 ring oligomers. Given that the polybasic patch and the $Ca^{2+}$-binding sites on C2B are located quite far apart, we suggest that polyphosphate binding rigidifies the C2B fold thereby better structuring the surfaces engaging in oligomerization. This long-range conformational coupling could explain the cooperative binding of PIP2 and $Ca^{2+}$ that has been reported for several C2 domain proteins, including Syt1 (*Montaville et al., 2008*; *Torrecillas et al., 2004*; *van den Bogaart et al., 2012*).

Our data raise the interesting possibility that under physiological conditions, the binding of cytosolic ATP to the C2B polybasic motif could initiate ring formation on the SV well before the docking forming a 'halo'-like structure (*Figure 5*). The average size of the ring oligomers observed by EM and FCS analysis suggests that the nearly all Syt1 molecule in the SVs may be involved, but the precise number of the Syt1 involved is unclear. Pre-formed Syt ring oligomers could then be transferred to PIP2 clusters at the PM to facilitate docking by virtue of the far greater affinity of a multi-valent Syt oligomer. This would require PM-bound PIP2 displacing free ATP as a ligand to the polybasic

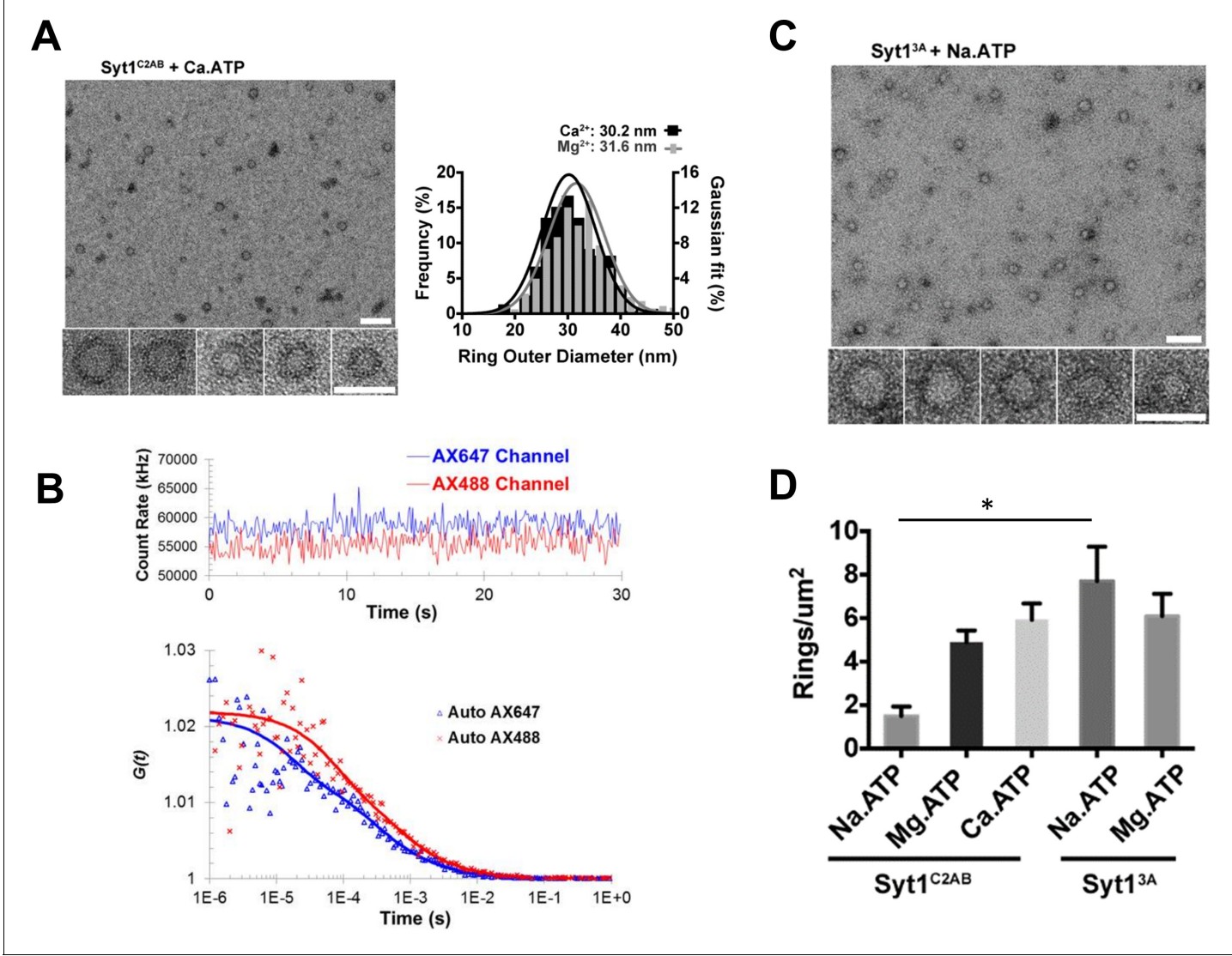

**Figure 4.** Soluble Syt1$^{C2AB}$ oligomers are insensitive to Ca$^{2+}$. (**A**) Negative stain analysis shows that the Syt1$^{C2AB}$ assembles into ring-like oligomers in the presence of Ca$^{2+}$ and ATP and these ring oligomers are similar in size and frequency to the rings assembled with Mg$^{2+}$/ATP. (**B**) Fluorescence autocorrelation analysis corroborates the insensitivity of the ring oligomers to Ca$^{2+}$ as the auto correlation function of Syt1$^{C2AB}$ incubated with Ca$^{2+}$ and ATP is best described with the two-component translational diffusion model with triplet state revealing the presence of the oligomeric component. (**C**) Neutralizing the Ca$^{2+}$ coordinating aspartic acids in the C2B domain stabilizes the soluble ring oligomers even in absence of Mg$^{2+}$ ions clarifying the molecular role of Mg$^{2+}$ in the soluble Syt1$^{C2AB}$ ring assembly. The stability of the ring oligomers under various conditions was quantified by density of the rings on the EM grid (**D**). Average values and deviations from a minimum of 3 independent experiments are shown.
DOI: https://doi.org/10.7554/eLife.27441.012

The following figure supplements are available for figure 4:

**Figure supplement 1.** Negative stain EM analysis shows that both Mg$^{2+}$ and Ca$^{2+}$ act in a similar manner to stabilize the soluble Syt1$^{C2AB}$ ring-like oligomers.
DOI: https://doi.org/10.7554/eLife.27441.013

**Figure supplement 2.** Fitting residues from fitting autocorrelation function for data shown in *Figure 3C*.
DOI: https://doi.org/10.7554/eLife.27441.014

patch, as previously demonstrated (*Park et al., 2012*; *Vennekate et al., 2012*). It is noteworthy that in the same study, it was found that ATP is a better ligand than PS on the SV surface, and suggested that ATP binding to C2B serves to prevent *cis*-binding of C2B to the SV thereby favoring ultimate *trans*-binding to PIP2 in the PM (*Park et al., 2012*; *Vennekate et al., 2012*). In accordance, we find

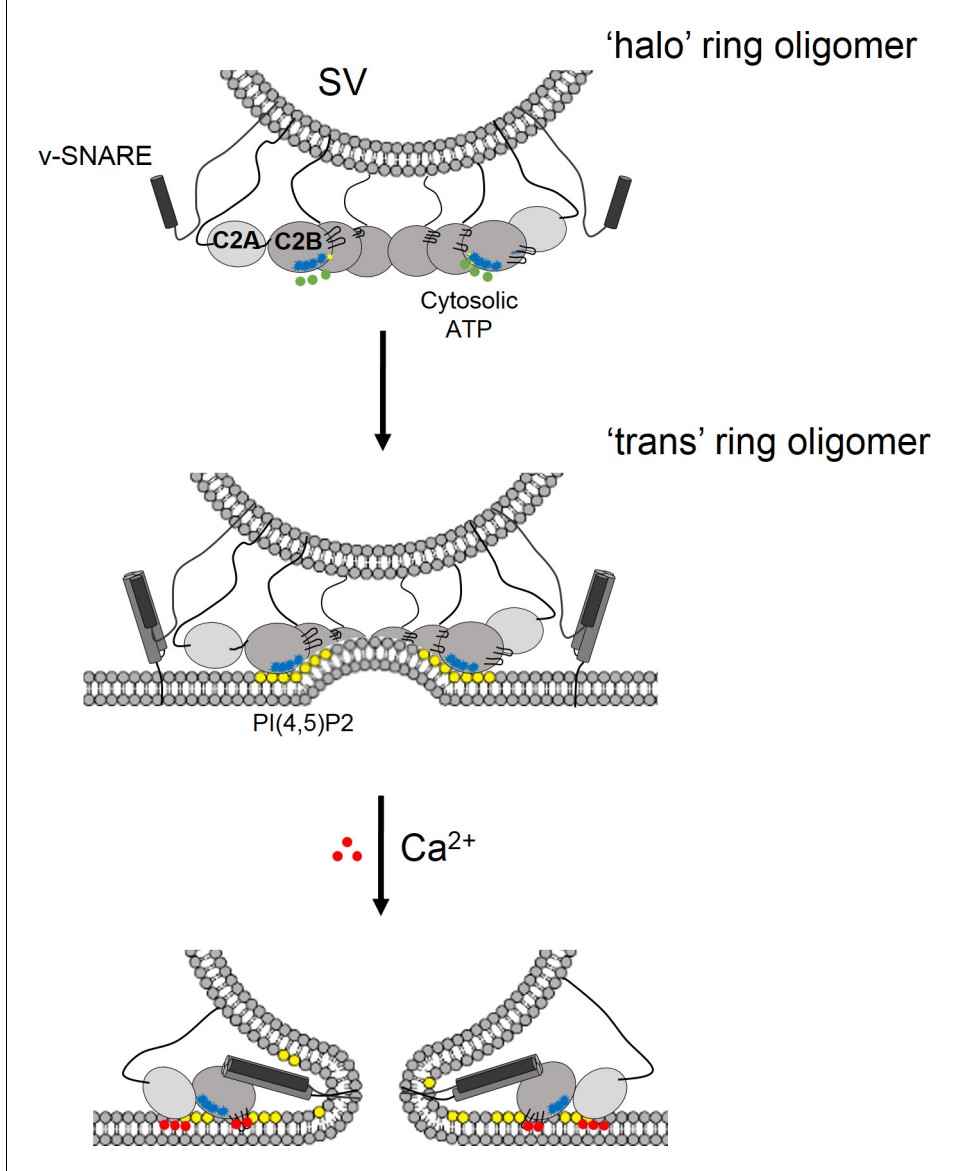

**Figure 5.** Model for the regulation of neurotransmitter release by the Syt1 ring oligomers. The Syt1 ring oligomer formation may be triggered prior to vesicle docking that is halo-like structure by cytosolic ATP (green dots) binding to the Syt1 C2B polylysine motif (blue dots). The ring oligomers are then transferred to the site of docking wherein the ATP interaction is replaced with the PIP2 clusters (yellow dots) on the plasma membrane. This serves to enhance the docking ability of the Syt1 and position the ring at the docking site prior to the engagement of the SNARE proteins. The dimensions of the Syt1 ring, would permit the assembly of N-terminal domain of SNARE, but impede the complete zippering either passively as a spacer/washer or actively by restraining the SNARE assembly via specific interaction with t-SNAREs. Upon binding $Ca^{2+}$ (red dots), the $Ca^{2+}$ loops that is at the oligomeric interface, re-orients and inserts into the membrane. This disrupts the Syt1 ring oligomer and removing the impediment for fusion. In this fashion, the Syt1 ring oligomers could synchronize the release neurotransmitters to the influx of $Ca^{2+}$ following the action potential.
DOI: https://doi.org/10.7554/eLife.27441.015

that the ATP drastically reduces the density of the Syt1 ring oligomers on lipid surface with PS only, but had no effect when PIP2 was included (*Zanetti et al., 2016*). Here, we extend this concept to suggest that this hierarchy of binding affinities may serve to pre-form a Syt1 ring as an additional mechanism to facilitate SV docking to the PM by Syt1.

The formation of the Syt1 ring oligomer prior to the engagement of SNAREs would allow it to generate and maintain the primed pre-fusion state. The dimensions of a Syt1 ring (diameter ~30 nm and height ~4 nm) considering 15–20 copies of Syt1 in the SVs (*Takamori et al., 2006*; *Wilhelm et al., 2014*) would permit the assembly of half zippered SNAREpins but could impede the complete zippering. It is not clear if Syt1 rings function to clamp fusion passively, as a spacer, to separate the two bilayers, or actively by physically restraining SNARE assembly via direct binding to the single SNARE complexes (*Choi et al., 2010*; *Zhou et al., 2015*). However, it is worth noting in our Syt1 ring oligomer model, the recently defined SNARE binding interface on the C2B (*Zanetti et al., 2016*) is accessible and free to interact in an active mechanism. Recent cryo-tomography studies of synaptosomes show that docked vesicles are ~3–4 nm away from plasma membrane with protein density at the interface. However, it does not allow for unambiguous identification of the protein(s) identity or their organization at the docked site. (*FernandezFernández-Busnadiego et al., 2011*).

Besides stabilizing the pre-fusion state, the Syt1 ring oligomer could also synchronize the release of neurotransmitter to $Ca^{2+}$ influx. $Ca^{2+}$ binding would trigger a rapid disintegration of the Syt1 oligomeric structure to permit the stalled SNAREpins to complete zippering and drive SV fusion. However, $Ca^{2+}$ sensitivity is not an innate property of the Syt1 oligomers, but requires the presence of lipid membranes. This reinforces our earlier finding that the $Ca^{2+}$-dependent interaction of the C2B domain with the membrane is key determinant in the ring disassembly (*Zanetti et al., 2016*). $Ca^{2+}$-induced conformational reorientation of the C2B loops into the membrane, which is the power stroke and physiologically required for triggering synaptic transmission (*Fernández-Chacón et al., 2001*; *Rhee et al., 2005*; *Paddock et al., 2011*), is incompatible with the ring-like structure, as it would disrupt the oligomeric interface. Overall, it is now easy to imagine how Syt1 oligomers could serve as template to organize assembling SNAREpins and related chaperones to enable the cooperative and synchronous triggering of neurotransmitter release. However, further research is needed to test the physiological relevance of these concepts.

# Materials and methods

Adenosine phosphates (ATP, ADP) and Inositol hexaphosphate (IP6) were purchased from Sigma-Aldrich (St Louis, MO). Inositol 1,4,5-trisphosphate (IP3), phosphatidylinositol 4,5-bisphosphate diC4 (PIP2-diC4) were purchased from Echelon biosciences (Salt Lake City, UT). 1-palmitoyl-2-oleoyl-sn-glycero-3-phosphocholine (POPC) and 1,2-dioleoyl-sn-glycero-3-phospho-L-serine (DOPS) were purchased from Avanti Polar lipids (Alabaster, AL). Thiol reactive fluorescent probes Alexa488-maleimide and Alexa647-maleimide were purchased from Thermo Scientific, Waltham, MA. The DNA constructs used in this study - the wild-type C2AB domain (Syt1$^{C2AB}$, residues 143–421) of rat Synaptotagmin-1, C2B polylysine mutant (Syt1$^{C2AB}$K326A, K327A), C2B $Ca^{2+}$ binding mutant (Syt1$^{3A}$, D309A, D363A, D365A) were generated and sequenced in our earlier works (*Wang et al., 2014*; *Zanetti et al., 2016*). For site-specific labeling with fluorophores, cysteine was introduced in Syt1-$^{C2AB}$ at residue 269, while naturally existing cysteine at residue 277 was removed (C$^{277}$S) using the Quickchange mutagenesis kit (Stratagene, Santa Clara, CA)

## Purification of recombinant proteins

The Syt1$^{C2AB}$ wild-type and mutant proteins were expressed and purified as a GST-tagged protein using a pGEX6 vector. The proteins were purified as described previously (*Wang et al., 2014*; *Zanetti et al., 2016*). Briefly, Escherichia coli BL21 (DE3) expressing Syt constructs were grown to an $OD_{600}$ ~0.7–0.8, induced with 0.5 mM isopropyl β-D-1-thiogalactopyranoside (IPTG). The cells were harvested after 3 hr at 37°C and suspended in lysis buffer (25 mM HEPES, pH 7.4, 400 mM KCl, 1 mM MgCl2, 0.5 mM TCEP, 4% Triton X-100, protease inhibitors). The samples were lysed using cell disrupter, and the lysate was supplemented with 0.1% polyethylamine before being clarified by centrifugation (100,000 × g for 30 min). The supernatant was loaded onto glutathione-sepharose (Thermo Scientific, Waltham, MA) beads (3 hr or overnight at 4°C), and the beads were washed with 20 ml of lysis buffer, followed by 20 ml of 25 mM HEPES, 400 mM KCl buffer containing with 2 mM ATP, 10 mM MgSO4, 1 mM DTT. Subsequently, the beads were re-suspended in 5 ml of lysis buffer supplemented with 10 µg/mL DNaseI, 10 µg/mL RNaseA, and 10 µl of benzonase (2000 units) and incubated at room temperature for 1 hr, followed by quick rinse with 10 ml of high salt buffer (25 mM HEPES, 1.1 M KCl, 1 mM DTT) to remove the nucleotide contamination. The beads were then

washed with 20 ml of HEPES, 400 mM KCl buffer containing 0.5 mM EGTA to remove any trace calcium ions. The proteins were eluted off the affinity beads in 25 mM HEPES, 100 mM KCl, 1 mM DTT buffer using PreScission protease for GST-tagged constructs and further purified by Mono-S anionic exchange (GE Healthcare, Marlborough, MA) chromatography. Size-exclusion chromatography (Superdex75 10/300 GL) showed a single elution peak (~13 mL) consistent with a pure protein, devoid of any contaminants.

## Imaging of soluble Syt1 ring oligomers

Protein stock (50 µM) was diluted for 10 fold in MBS (20 mM MOPS, pH7.5, 100 mM KCl, 1 mM EGTA, 1 mM Mg(AC)2, 1 mM Mg.ATP, 1 mM DTT, 4% trehalose) at room temperature for 10 min. Alternatively, Mg.ATP can be substituted with 1 mM Mg.ADP, 50 uM IP3, 50 uM PIP2-diC4, 1 mM Ca.ATP. Diluted protein solution is further centrifuged at 10,000 x g for 10 min at 4°C to remove large aggregates and the supernatant (~8 µl) was applied to a continuous carbon-coated EM grid, which were glow-discharged for 10 s prior to application. After 1 min incubation, the grid was blotted dry with Whatman #1 filter paper, stained with 1% uranyl acetate and air-dried. The negatively stained specimens were examined on a transmission electron microscope (FEI Tecnai T12) operated at an acceleration voltage of 120 keV. The defocus range used for our data was from 0.6 to 2.0 um. Images were recorded under low-dose conditions (~20 e⁻/Å2) on a 4K × 4K CCD camera (UltraScan 4000; Gatan, Inc), at a nominal magnification of 42,000x. Micrographs were binned by a factor of 2 at a final sampling of 5.6 Å per pixel on the object scale. A minimum of three independent analyses were used for each condition and average and standard error of the means (SEM) from this dataset is shown. Typically, randomly selected 20–30 regions of 500 × 500 nm dimensions from each individual trial (a minimum of 100 regions) was used for the density analysis and a minimum of 200 ring structures were used to estimate the size distribution.

## Soluble ring purification

An aliquot of 10 µl Syt1$^{C2AB}$ stock was diluted in 90 µl MBS. After 10 min incubation at room temperature with ATP and Mg$^{2+}$, the protein solution mixed with 10 µl of buffer or buffer containing 0.01% glutaraldehyde on ice for 10 min and then laid on top of a glycerol density gradient (100 µl layers of 10%, 15%, 20%, 25%, 30% glycerol (v/v), which were allowed to settle for 1 hr at 4°C) and centrifuged at 100,000 x g for 16 hr at 4°C. After the centrifugation, the samples were collected as 50 µl aliquots (from top to the bottom) and directly imaged for negative staining EM. To increase the concentration of particles on the EM grid, samples were allowed to adsorb on the carbon surface for 1 hr in a humidity chamber on ice. An alternate sample under the same conditions except without the fixative was tested in parallel.

## Fluorescence Correlation Spectroscopy analysis

Fluorescence correlation spectroscopy experiments were performed using a Carl Zeiss LSM 510 confocal microscope with a correlator module. A 40x water immersion objective lens was used in all experiments. All experiments were carried out with total (labeled + unlabeled) Syt1$^{C2AB}$ concentration of 5 µM of Syt1C2AB in 25 mM HEPES, 20 mM KCl, 1 mM Mg$^{2+}$ buffer with or without 1 mM Mg.ATP included. The concentration of Alexa488/Alexa647 labeled Syt1$^{C2AB}$ were adjusted as per the experimental requirement. For cross correlation measurements, typically ~120 nM of Syt1$^{C2AB}$-Alexa488, ~30–120 nM of Syt1$^{C2AB}$-Alexa647 were mixed with ~4.9 µM of unlabeled Syt1$^{C2AB}$ at room temperature for 15 min. The sample was then loaded into a glass bottom microwell dish (MatTek Corporation, Ashland, MA) for imaging. Under some cases, the sample was then rapidly diluted (5x) with buffer and immediately subjected to imaging. For auto correlation experiments, concentrations of each fluorescent species lowered to ~10–60 nM of Syt1$^{C2AB}$-Alexa488, ~10–60 nM of Syt1-$^{C2AB}$-Alexa647 and ~5 µM of unlabeled Syt1$^{C2AB}$. Typically, the sample was pre-mixed in eppendorf tubes, and then transferred into a glass bottom microwell dish (MatTek Corporation, Ashland, MA) for imaging. Both fluorophores were directly excited using 488 nm and 633 nm lasers.

The auto-correlation function is defined as follows:

$$G(\tau) = \frac{\langle I(t).I(t+\tau)\rangle}{\langle I(t)\rangle^2} = \frac{T\int_0^T (I(t).I(t+\tau))dt}{\int_0^T (I(t))^2 dt}$$

where $I(t)$ is the fluorescence intensity at time $t$, and $<>$denotes the time average, and $\tau$ is the time shift.

The definition for the cross correlation function is identical to the auto correlation function, with the exception that the signal in one channel is compared to a signal in a second channel instead of itself.

$$G_X(\tau) = \frac{\langle I_B(t).I_Y(t+\tau)\rangle}{\langle I_B(t).I_Y(t)\rangle}$$

where the indices 'B' and 'Y' refer to the red and blue channel, respectively.

The total correlation is given by the following equation:

$$G_{tot}(\tau) = 1 + d + C + A.k\prod\sum_l G_{k,l}(\tau)$$

where $d$ is the offset, $C$ is the background correction, and $A$ is the amplitude. $G_{k,\,l}(\tau)$ is the correlation for a single process. The suffixes $k$ and $l$ are correlation terms for dependent and independent processes, respectively, that are multiplied with or added to each other.

For a system with a single species of translationally diffusional particles, the above equation can be simplified as

$$G(\tau) = 1 + A\left(1 + \frac{\Phi_{tri}e^{-\frac{\tau}{\tau_{tri}}}}{1 - \Phi_{tri}}\right)\frac{1}{\left(1 + \frac{\tau}{\tau_d}\right)\left(1 + \frac{1}{s^2}\frac{\tau}{\tau_d}\right)^{1/2}}$$

where $\Phi_{tri}$ and $\tau_{tri}$ are the fraction and decay time of the triplet state. $\tau_d$ is the diffusional time of the particle, and $s$ is the structural parameter.

If there are two species of particles diffusing in the focal volume, the correlation function can be written as

$$G(\tau) = 1 + A\left(1 + \frac{\Phi_{tri}e^{-\frac{\tau}{\tau_{tri}}}}{1 - \Phi_{tri}}\right)\left(\frac{\Phi_1}{\left(1 + \frac{\tau}{\tau_1}\right)\left(1 + \frac{1}{s^2}\frac{\tau}{\tau_1}\right)^{1/2}} + \frac{\Phi_2}{\left(1 + \frac{\tau}{\tau_2}\right)\left(1 + \frac{1}{s^2}\frac{\tau}{\tau_2}\right)^{1/2}}\right)$$

where $\Phi_1$ and $\tau_1$ are the fraction and diffusional time of the component 1, $\Phi_2$ and $\tau_2$ are the fraction and diffusional time of the component 2. These equations can be used to determine the diffusional properties of the particles. Fitting residues from fitting autocorrelation function for all FCS data are shown in *Figure 4—figure supplement 2*.

## Acknowledgements

This work was supported by the National Institute of Health grants DK027044 and GM118084 to JER. We acknowledge the use of CCMI Electron Microscopy Core Facility, supported by Yale School of Medicine.

## Additional information

### Funding

| Funder | Grant reference number | Author |
| --- | --- | --- |
| National Institute of Diabetes and Digestive and Kidney Diseases | DK027044 | James E Rothman |

| National Institute of General Medical Sciences | GM118084 | James E Rothman |
|---|---|---|

The funders had no role in study design, data collection and interpretation, or the decision to submit the work for publication.

### Author contributions
Jing Wang, Feng Li, Conceptualization, Data curation, Formal analysis, Investigation, Writing—original draft, Writing—review and editing; Oscar D Bello, Conceptualization, Investigation, Writing—review and editing; Charles Vaughn Sindelar, Conceptualization, Formal analysis, Methodology, Writing—review and editing; Frédéric Pincet, Conceptualization, Formal analysis, Supervision, Methodology, Writing—review and editing; Shyam S Krishnakumar, James E Rothman, Conceptualization, Formal analysis, Supervision, Funding acquisition, Writing—original draft, Writing—review and editing

### Author ORCIDs
Charles Vaughn Sindelar http://orcid.org/0000-0002-6646-7776
Shyam S Krishnakumar https://orcid.org/0000-0001-6148-3251
James E Rothman http://orcid.org/0000-0001-8653-8650

### Decision letter and Author response
Decision letter https://doi.org/10.7554/eLife.27441.018
Author response https://doi.org/10.7554/eLife.27441.019

## Additional files

### Supplementary files
• Transparent reporting form
DOI: https://doi.org/10.7554/eLife.27441.016

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
