## [Decision Letter]

Thank you for submitting your article "Circular Oligomerization is an Intrinsic Property of Synaptotagmin" for consideration by *eLife*. Your article has been reviewed by three peer reviewers, and the evaluation has been overseen by a Reviewing Editor and Randy Schekman as the Senior Editor. The reviewers have opted to remain anonymous.

The reviewers have discussed the reviews with one another and the Reviewing Editor has drafted this decision to help you prepare a revised submission.

Summary:

In this manuscript, the authors describe follow-up experiments that are related to their 2016 *eLife* paper. They observed that the C2AB fragment of syt1 is able to form circular oligomers in solution. The ring-like structures of syt1 C2AB require PIP2 analogues, ADP, or ATP as well as Ca^2+^ or Mg^2+^ ions. The authors suggest a model in which the syt1 ring oligomers serve to synchronize the release neurotransmitters upon Ca^2+^ influx. The rings are interesting, but it seems their biological function remains unknown, and such rings have not (yet) been observed in neurons.

Essential revisions:

1) The authors show that both Ca^2+^ and Mg^2+^ can efficiently promote formation of the ring-like structures. It is well established that syt1 binds to Ca^2+^ is in the microM range (e.g., A. Radhakrishnan, A. Stein, R. Jahn, D. Fasshauer, J. Biol. Chem. 284, 25749-60, 2009). However, the affinity of Mg^2+^ to syt1 has not been well characterized. In fact, the concentration of Mg^2+^ in the cytosol is nearly 1 mM, so Ca^2+^, so Ca^2+^ would compete with Mg^2+^ at lower concentration. Are Mg^2+^ and Ca^2+^ acting in the same manner, to drive ring formation? Is there any difference in the abilities of these two ions to drive ring formation? Probing interactions between syt1 and divalent cations by ITC and testing ring-like structure formation as a function of divalent cation concentration would be necessary to address this. Also, is the ring-like structure formed in the presence of other ions as well?

2) Does PIP2 alone help to form the oligomers? Since the authors showed that ATP alone cannot lead to the ring-like structure, it would be interesting to see if PIP2 alone gives the same result.

3) As the oligomers can only be observed after fixation (0.01% glutaraldehyde), are they really formed inside cells? In addition, full-length syt1 is anchored on the membrane of synaptic vesicles (SV, ~ 40 nm) through its N-terminal transmembrane domain. If it also forms 30 nm rings, as does the soluble domain (C2AB), it seems it should be readily observable by EM. However, even with the most advanced electron cryo-tomography studies such rings have not been observed (e.g., R. Fernández-Busnadiego et al., J. Cell Biol. 201, 725-740, 2013). The authors should speculate why such rings have not (yet) been observed in vivo.

4) In Figure 7D in Xue et al. (2010) Nature SMB 17(5): 568 it has been reported that there is a normal vesicle release probability (Pvr) in syt1 -/+ versus syt1 +/+ in mouse neurons. This observation seems to be at odds with the suggestion that all Syt1 molecules in particular synaptic vesicle are used for a possible ring since there probably would be a smaller number of Syt1 molecules in Syt1 -/+ mice. Please comment.

5) The FCCS experiments showed that 20% of C2AB forms oligomers in solution without 0.01% glutaraldehyde. Why is it that this 20% cannot be observed by glycerol density gradient analysis and/or by negative stain EM, at all?

6) In the Figure 3, it seems that the concentrations of the fluorophores (Alexa488-Syt1 and Alexa 647-Syt1) between the two groups are not similar (the G(0) values displayed the concentration used in Figure 3 is much smaller than that in Figure 3). Although it will not affect the final FCCS results (still at permissive concentration, actually), the fits will be disturbed in Figure 3 at lower time scale so that one can imagine that there will be large errors of the parameters acquired by fitting to ACF.

7) In the Figure 4, the numbers of the rings of Syt13A are larger than Syt1C2AB when applied to Na-ATP. Is this statistical significant?

8) In the discussion of Figure 5, the authors suggest that "The dimensions of the Syt1 ring,.…by restraining the SNARE assembly via specific interaction with t-SNAREs. " in the legend. However, why did the authors not depict the interaction between Syt1 and t-SNAREs in the "'trans ring oligomer"?

Optional revisions:

9) With the detailed biophysical measurements, it is possible to map conserved motifs mediating C2AB oligomerization. Apart from the K326A/K327A mutations in the polybasic region, identification of residues that form the other part of the interface would strengthen the manuscript. In other words, which region does the polybasic region interact with?

10) It would be interesting to test Doc2b, which intrinsically lacks a transmembrane anchor, and compare the data with C2AB results.

11) The authors stated that "No oligomeric structures were observed in the absence of polyvalent anions and the density of the rings was greatly reduced from (~4 rings/µm 2 to ~0.8 rings/µm 2) when the polybasic region of C2B was mutated (K326A/K327A) (Figure 2)". But this it not shown in Figure 2.

12) The authors are encouraged to provide the fitting residues for all of the FCS/FCCS results.

13) In Figure 1, the authors show that the diameters of the oligomers are slightly different in the presence (~28 nm) or absence (~32 nm) of lipid monolayers. The authors are encouraged to interpret the results in the main text or in the Discussion section (i.e., what it stands for? Is there any physiological significance?).

---

## [Author Response]

*Essential revisions:*

*1) The authors show that both Ca^2+^ and Mg^2+^ can efficiently promote formation of the ring-like structures. It is well established that syt1 binds to Ca^2+^ is in the microM range (e.g., A. Radhakrishnan, A. Stein, R. Jahn, D. Fasshauer, J. Biol. Chem. 284, 25749-60, 2009). However, the affinity of Mg^2+^ to syt1 has not been well characterized. In fact, the concentration of Mg^2+^ in the cytosol is nearly 1 mM, so Ca^2+^, so Ca^2+^ would compete with Mg^2+^ at lower concentration. Are Mg^2+^ and Ca^2+^ acting in the same manner, to drive ring formation? Is there any difference in the abilities of these two ions to drive ring formation? Probing interactions between syt1 and divalent cations by ITC and testing ring-like structure formation as a function of divalent cation concentration would be necessary to address this. Also, is the ring-like structure formed in the presence of other ions as well?*

Our data suggests that both Mg^2+^ and Ca^2+^ act in a similar manner to stabilize the soluble ring structures. It does so by neutralizing the Asp residues in the Ca^2+^-binding loops, which likely locate at the C2B-C2B interface in the oligomer. Binding of Syt1 to Ca^2+^ and Mg^2+^ quantified by MST analysis (van den Bogaart et al., 2012) shows that Syt1 has very low, non-determinable affinity to Mg^2+^ as compared to Ca^2+^ (K_d_ ~250 µM). Nevertheless, Mg^2+^ has been known to coordinate the Asp residues and in fact, has been observed in the X-ray structure (Zhou et al., 2015). Consistent with this, we find both Mg^2+^ & Ca^2+^ can stabilize the oligomeric structures across range of concentration (0.25 – 1 mM), but Ca^2+^ is better across the whole range. This information is now discussed in Results section and included as Figure 4—figure supplement 1. We speculate all divalent cations might act in the same way, albeit the stabilizing effect might vary depending on their affinity to Syt1.

*2) Does PIP2 alone help to form the oligomers? Since the authors showed that ATP alone cannot lead to the ring-like structure, it would be interesting to see if PIP2 alone gives the same result.*

Upon careful review of our data, we believe that PIP2 or ATP alone trigger the assembly of the Syt1 ring oligomers under soluble conditions and with comparable efficacy. However, the density of the ring oligomers is low. Mg^2+^ (and Ca^2+^) increase the number of these oligomeric structures observed (~1.5x), likely by stabilizing the oligomeric form. This is evident in the direct comparison of the ring oligomers formed with PIP2 & ATP in the presence and absence of Mg^2+^. In accordance, we have updated Figure 2 and have also re-worded the Results section as follows: “Mg^2+^ was also found to be an important co-factor in stabilizing the Syt1^C2AB^ rings in solution (Figure 2). The number of Syt1^C2AB^ ring oligomers observed were substantially reduced (~60% reduction) when Mg^2+^ was completely excluded (using Na.ATP or Na.PIP2) (Figure 2). Given that the Mg^2+^ was not required to assemble the Syt1^C2AB^ ring oligomers on acidic lipid surfaces (in fact Mg^2+^ lowered the number of rings formed on lipid surfaces), it appears that Mg^2+^ plays an independent but auxiliary role in the formation of Syt1^C2AB^ ring oligomers in solution”.

*3) As the oligomers can only be observed after fixation (0.01% glutaraldehyde), are they really formed inside cells? In addition, full-length syt1 is anchored on the membrane of synaptic vesicles (SV, ~ 40 nm) through its N-terminal transmembrane domain. If it also forms 30 nm rings, as does the soluble domain (C2AB), it seems it should be readily observable by EM. However, even with the most advanced electron cryo-tomography studies such rings have not been observed (e.g., R. Fernández-Busnadiego et al., J. Cell Biol. 201, 725-740, 2013). The authors should speculate why such rings have not (yet) been observed* in vivo.

Syt1 ring oligomers are observed under native conditions and does not require fixative for visualization by negative stain EM. A very small amount of fixative (0.01% glutaraldehyde) is ONLY needed when we attempted to isolate these oligomers by glycerol gradient (see Question 5 for detailed response regarding this). Our data indicates that the Syt1 rings can assemble under physiological ionic strength and membrane composition. Given the length/flexibility of the juxtamembrane linker region, it should not affect the C2 ring formation, even when anchored in the synaptic vesicle. In fact, mathematical modeling, under physiological conditions and considering the vesicle dimension and the physical constraints, faithfully reproduces the Syt1 oligomerization in solution and it’s subsequently localization to the vesicle-plasma membrane interface. A manuscript describing these findings is currently under consideration (Zhu, O’Shaughnessy & Rothman).

Cryo-tomography studies (R. Fernández-Busnadiego et al. J. Cell Biol., 2010 & 2013) of the synaptosomal fractions find that the docked vesicles are typically ~ 3-4 nm away from the plasma membrane (consistent with the height of the ring oligomers) and show protein density at the vesicle-plasma membrane interface. However, the relatively low resolution of these tomograms, combined with the crowded environment of the synaptosomes, does not allow for unambiguous identification of the protein(s) identity or their organization at the docked site. This is further compounded by the fact that tomograms only show the side view of the Syt1 oligomers and dimensions at the rims of ~ 3-5 nm (considering relative orientations of C2A & C2B) might not be resolvable under these conditions.

It is also possible that under the experimental conditions used for the synaptosomal preparation which involves low temperature, multiple rounds of centrifugation and percoll gradient, with low ATP concentration and/or the freezing process might have disrupted or destroyed the Syt1 ring oligomers, as C2B-C2B ring oligomer interaction is not very strong. In fact, we require mild-fixative to visualize the isolated oligomers following high-speed centrifugation process. We currently working on visualization of the Syt1 oligomers under in vivo conditions using both super-resolution microscopy and cryo-electron microscopy. However, this is beyond the scope the current paper.

We have now included the following statement in the Discussion section to address this: “Recent cryo-tomography studies of synaptosomes show that docked vesicles are ~3-4 nm away from plasma membrane with protein density at the interface. However, it does not allow for unambiguous identification of the protein(s) identity or their organization at the docked site. (Fernandez-Busnadiego et al., 2011).”

*4) In Figure 7D in Xue et al. (2010) Nature SMB 17(5): 568 it has been reported that there is a normal vesicle release probability (Pvr) in syt1 -/+ versus syt1 +/+ in mouse neurons. This observation seems to be at odds with the suggestion that all Syt1 molecules in particular synaptic vesicle are used for a possible ring since there probably would be a smaller number of Syt1 molecules in Syt1 -/+ mice. Please comment.*

This is most likely due to the bottle neck with protein trafficking to axons/boutons. If this is indeed the limiting stage, the amount of Syt1 expressed in the soma is not really important (within a reason). One should expect similar levels of Syt1 in synaptic vesicles in the axonal boutons in Syt1 +/+ and +/- mice whilst the somatic expression may indeed be lower in +/- mice. Further, given the flexibility in the size of the ring oligomers, minor alterations in the Syt1 density can be accommodated in our model. We have included following clarification in the Discussion section “The average size of the ring oligomers observed by EM and FCS analysis suggest that the nearly all Syt1 molecule in the SVs may be involved, but the precise number of the Syt1 involved is unclear”

*5) The FCCS experiments showed that 20% of C2AB forms oligomers in solution without 0.01% glutaraldehyde. Why is it that this 20% cannot be observed by glycerol density gradient analysis and/or by negative stain EM, at all?*

It is likely that the fractionation of Syt1 in the glycerol gradient results in dilution of the Syt1 concentration below the threshold for oligomer formation. Further, the high speed centrifugation process might disrupt the ring oligomers. Inclusion of very small amount of fixative (0.01% glutaraldehyde) serves to stabilize these pre-formed rings and enable their visualization.

*6) In the Figure 3, it seems that the concentrations of the fluorophores (Alexa488-Syt1 and Alexa 647-Syt1) between the two groups are not similar (the G(0) values displayed the concentration used in Figure 3 is much smaller than that in Figure 3). Although it will not affect the final FCCS results (still at permissive concentration, actually), the fits will be disturbed in Figure 3 at lower time scale so that one can imagine that there will be large errors of the parameters acquired by fitting to ACF.*

It is true that the concentration of the sample shown in Figure 3 (presence of Mg^2+^/ATP) is lower than that shown in Figure 3 (absence of Mg^2+^/ATP). We performed the correlation measurements using a range of concentrations, but specifically chose to present the data in this format. Our intention was to highlight the fact that we observe no cross-correlation even at higher label concentration in the absence Mg^2+^/ATP in contrast to distinct cross-correlation in the presence of Mg^2+^/ATP at lower label concentration. We have now made this explicit in the Figure 3 legend.

As for the ACF curves, we agree with the reviewer that the parameters obtained from fitting high yet permissive concentration samples are less accurate. Hence, we always fitted for a range of concentration and increased the number of independent measurements for higher concentration to minimize the errors such that the averaged parameters would be more accurate.

*7) In the Figure 4, the numbers of the rings of Syt13A are larger than Syt1C2AB when applied to Na-ATP. Is this statistical significant?*

Yes. The number of rings observed with Syt1^3A^with Na.ATP is statistically significant than the number of Syt1^C2AB^rings under the same conditions. This is not indicated in the Figure 4.

*8) In the discussion of Figure 5, the authors suggest that "The dimensions of the Syt1 ring,.…by restraining the SNARE assembly via specific interaction with t-SNAREs. " in the legend. However, why did the authors not depict the interaction between Syt1 and t-SNAREs in the "'trans ring oligomer"?*

As it is stated in figure legend – It is not clear if the Syt1 ring blocks fusion passively as a washer/spacer to keep the membrane apart (or) actively by restraining the SNARE complex assembly. Further, the positioning of SNAREs with respect with the Syt1 ring is also not known. For these reasons, we chose not to depict the interaction of SNAREs with the Syt1 ring in Figure 5. This is an active area of current research.

*Optional revisions:*

*9) With the detailed biophysical measurements, it is possible to map conserved motifs mediating C2AB oligomerization. Apart from the K326A/K327A mutations in the polybasic region, identification of residues that form the other part of the interface would strengthen the manuscript. In other words, which region does the polybasic region interact with?*

Based on the 3D reconstruction of the Syt1 tubes, we have in fact mapped the C2B oligomerization region and have identified several Syt1 ring altering mutations. We are currently testing these mutations using in vitro and in vivofunctional analysis to establish the physiological role of the Syt1 rings. We are preparing a manuscript describing these findings currently.

*10) It would be interesting to test Doc2b, which intrinsically lacks a transmembrane anchor, and compare the data with C2AB results.*

We are currently carrying out comprehensive analysis of other C2 domain proteins, including other isoforms of Synaptotagmin and Doc2B. We are also extending this analysis to look at hetero-oligomerization between different isoforms.

*11) The authors stated that "No oligomeric structures were observed in the absence of polyvalent anions and the density of the rings was greatly reduced from (~4 rings/µm 2 to ~0.8 rings/µm 2) when the polybasic region of C2B was mutated (K326A/K327A) (Figure 2)". But this it not shown in Figure 2.*

The EM data for the K326A/K327A mutational analysis is shown in Figure 2 and the quantitation is shown in Figure 2. This has been corrected.

*12) The authors are encouraged to provide the fitting residues for all of the FCS/FCCS results.*

The fitting residuals for the FCS/FCCS results are now included in Figure 4—figure supplement 2.

*13) In Figure 1, the authors show that the diameters of the oligomers are slightly different in the presence (~28 nm) or absence (~32 nm) of lipid monolayers. The authors are encouraged to interpret the results in the main text or in the Discussion section (i.e., what it stands for? Is there any physiological significance?).*

We speculate that the deformation of the lipid monolayer, specifically the positive curvature induced by the Syt1^C2AB^ monomers, might make the rings more compact on membrane surface as compared to a more rigid, soluble structures. However, given the size distribution of the Syt1 ring oligomers and the dimensions of the Syt1 monomer, we do not believe that difference in the diameter difference between these two conditions (28 ± 2 nm vs. 32 ± 3 nm) is significant. The dimensions are in the range observed with the entire cytoplasmic domain of Syt1 and other Syt isoforms (28 – 31 nm). To clarify this, we have included the size distribution detail in Figure 1.